# Absence of Microglial Activation and Maintained Hippocampal Neurogenesis in a Transgenic Mouse Model of Crohn’s Disease

**DOI:** 10.3390/cells14110841

**Published:** 2025-06-04

**Authors:** Rebecca Katharina Masanetz, Hanna Mundlos, Iris Stolzer, Jürgen Winkler, Claudia Günther, Patrick Süß

**Affiliations:** 1Department of Molecular Neurology, Universitätsklinikum Erlangen, Friedrich-Alexander-Universität Erlangen-Nürnberg, 91054 Erlangen, Germany; 2Department of Medicine 1, Universitätsklinikum Erlangen, Friedrich-Alexander-Universität Erlangen-Nürnberg, 91054 Erlangen, Germany; 3Deutsches Zentrum Immuntherapie, Universitätsklinikum Erlangen, 91054 Erlangen, Germany; 4Profile Center Immunomedicine, Friedrich-Alexander-Universität Erlangen-Nürnberg, 91054 Erlangen, Germany

**Keywords:** adult hippocampal neurogenesis, dentate gyrus, microglia, gut–immune–brain axis, inflammatory bowel disease

## Abstract

Adult neurogenesis in the hippocampal dentate gyrus (DG) is not only essential for learning and pattern separation, but it is also involved in emotional regulation. This process is vulnerable to local and peripheral inflammation, which is partly mediated by microglia in the DG. As Crohn’s disease (CD) is associated with neuropsychiatric comorbidity, including depression and cognitive impairment, a reduction in adult hippocampal neurogenesis by chronic gut-derived inflammation has been hypothesized. Here, we present the first study that examined the influence of chronic ileocolitis on microglia in the DG and on adult hippocampal neurogenesis in a transgenic mouse model of CD, which is generated by a constitutive knockout of *caspase 8* in intestinal epithelial cells (IECs, *Casp8^ΔIEC^* mice). Structural and transcriptional analyses revealed that microglial cell proliferation and density in the DG as well as the expression of genes associated with their homeostasis and activation in the forebrain were maintained in 14- and 24-week-old *Casp8^ΔIEC^* mice compared to *Casp8^fl^* controls. Furthermore, different stages of adult hippocampal neurogenesis, including progenitor cell proliferation, maturation, and apoptosis of newly generated cells, were predominantly unaffected by chronic ileocolitis, except a potential minor phenotypic shift in maturating cells in 24-week-old mice. Together, we demonstrate largely preserved adult hippocampal neurogenesis, lacking signs of local inflammatory microglial activation despite chronic inflammation of the gut.

## 1. Introduction

Besides gastrointestinal symptoms, a high disease burden in inflammatory bowel diseases (IBDs), which comprise ulcerative colitis (UC) and Crohn’s Disease (CD), is conveyed by comorbid neuropsychiatric symptoms. Meta-analyses have discovered an increased prevalence of anxiety and depression, as well as cognitive impairment, in IBD patients [1,2,3]. Furthermore, anxiety and depression symptoms were even higher in patients with CD compared to UC patients [1]. The mechanisms underlying neuropsychiatric comorbidities are not fully understood, but increasing evidence suggests a bidirectional communication between the gut and the central nervous system (CNS) via the immune system, which may lead to the propagation of gut-derived inflammation into the brain (see for review [4]).

As a potential downstream effect, impaired adult hippocampal neurogenesis has been proposed to contribute to neuropsychiatric comorbidity in IBD. Adult hippocampal neurogenesis is defined as the generation of new neurons in the hippocampal dentate gyrus (DG) during adulthood, which is not only essential for learning, memory, and pattern separation but also significantly involved in emotional regulation [5]. Several trophic factors govern adult neurogenesis, including the brain-derived neurotrophic factor (BDNF), glial cell line-derived neurotrophic factor (GDNF), and insulin-like growth factor 1 (IGF1) [6,7,8]. Impairment of this highly vulnerable and plastic process is linked to cognitive impairment or affective disorders like anxiety and depression. In addition, a drug-mediated increase in adult hippocampal neurogenesis is able to reduce anxiety- and depression-related behaviors in mice [5,9,10]. Furthermore, it is well established that adult hippocampal neurogenesis is impaired by local neuroinflammation as well as peripheral inflammation, such as chronic complete Freund’s adjuvant-induced inflammatory arthritis and peripheral lipopolysaccharide-induced inflammation [11,12,13,14]. This detrimental effect of inflammation on adult hippocampal neurogenesis is substantially mediated by microglia, the resident macrophages of the brain [15,16]. Microglia are critical to maintain homeostasis in the adult hippocampal neurogenic niche through phagocytosis and chemokine release in healthy, aged, and diseased brains [12,16,17,18]. In the context of gut-derived peripheral inflammation, previous studies focused on the dextran sodium sulfate (DSS) chemically induced colitis mouse model, which led to conflicting results regarding effects on adult hippocampal neurogenesis (see [4] for review). So far, data on the effects of chronic ileocolitis on adult hippocampal neurogenesis in an animal model more specifically reflecting features of CD are missing.

In this study, we employed a genetically modified mouse model that spontaneously develops ileocolitis and symptoms analogous to those observed in CD, including a cohort of male and female adolescent (14-week-old) and adult (24-week-old) mice. The primary hypothesis was that chronic genetically induced colitis results in microglial activation in the DG associated with an impaired adult hippocampal neurogenesis. To the best of our knowledge, this is the first longitudinal study to investigate the impact of chronic colitis on adult hippocampal neurogenesis using a genetically modified mouse model that spontaneously develops ileocolitis in adolescent and adult mice.

## 2. Materials and Methods

### 2.1. Experimental Animals

All animal experiments were approved by the local government commission of animal health and were performed according to the EU Directive 2010/63 for the protection of animals used for scientific purposes. The animals were housed under specific-pathogen-free conditions in a 12 h light/dark cycle and provided with drinking water and standard laboratory chow ad libitum. Male and female experimental mice with an intestinal epithelial cell (IEC)-specific deletion of *caspase 8* (*Casp8^ΔIEC^*) and *Casp8^fl^* control mice were used in this study [19]. *Casp8^ΔIEC^* mice spontaneously develop intestinal inflammation typically around 12 weeks of age, influenced by the composition of the gut microbiota and environmental factors [20,21]. To investigate how different durations of ileocolitis affect the DG, we selected two time points: 14 weeks, corresponding to early inflammation, and 24 weeks, representing a more advanced and sustained inflammatory state. All experimental groups contained mixed genders.

### 2.2. Tissue Sample Preparation

Mice were deeply anesthetized using isofluorane and transcardially perfused with ice-cold phosphate-buffered saline (PBS). The brain was exposed, and one hemisphere was snap-frozen and stored at −80 °C until RNA isolation, while the contralateral hemisphere was fixed in 4% paraformaldehyde for 4–8 h and rehydrated in 30% sucrose solution before sagittal slicing into 20 µm sections with a sliding microtome (Leica SM2010 R, Leica, Nussloch, Germany). The colon and intestine were dissected and rinsed with PBS. Tissue sections of the distal colon, ileum, and liver were snap-frozen and stored at −80 °C until RNA isolation. Additionally, tissue sections of the distal colon, ileum, and liver were fixed in 4.5% formaldehyde, dehydrated, and embedded in paraffin before cutting into 3 µm slices.

### 2.3. Histological Analysis

Histopathological analysis of colitis and ileitis was performed on formalin-fixed paraffin-embedded tissue after hematoxylin and eosin (H&E) staining.

The staining of the brain tissue was performed in every 12th sagittal brain section of each animal. Nissl staining of sagittal brain slices was performed by hydrating the samples in ethanol at decreasing concentrations (100%, 95%, 70%, 50%), rinsing in distilled water, staining in thionin staining solution (0.25% thionin, 36 mM NaOH, 200 mM acetic acid in H_2_O), rinsing in water, rehydrating in ethanol at increasing concentrations (50%, 70%, 95%, 100%), and clearing in Neo-Clear^®^ (Merck, Darmstadt, Germany) before covering with Neo-Mount^®^ (Merck, Darmstadt, Germany).

Immunofluorescence staining of free-floating sagittal brain sections was performed as previously described [22]. The sections were washed three times in a permeabilization buffer (0.05% Triton™ X-100 in Tris-buffered saline (TBS)), incubated in a citrate buffer (0.1 M citric acid, 0.1 M Tris-sodium citrate in H_2_O) for 30 min at 80 °C, and washed three times in the permeabilization buffer prior to being transferred into a blocking solution (3% donkey serum, 0.3% Triton™ X-100 in TBS) for 2 h at RT. Following incubation with primary antibodies overnight at 4 °C, the sections were washed three times in the permeabilization buffer and incubated with secondary antibodies for 2 h at RT. The primary antibodies used were a rat monoclonal antibody against a cluster of differentiation of 68 (CD68; 1:300, Bio-Rad, Feldkirchen, Germany), a goat polyclonal antibody against ionized calcium-binding adapter molecule 1 (Iba1; 1:500, Abcam), a rabbit polyclonal antibody against minichromosome maintenance complex component 2 (Mcm2; 1:300, Cell Signaling Technology, Danvers, MA, USA), and a goat polyclonal antibody against doublecortin (DCX; 1:250, Santa Cruz Biotechnology, Dallas, TX, USA). Secondary antibodies used were fluorophore-conjugated (Alexa Fluor 488, Alexa Fluor 568, Alexa Fluor 647) donkey antibodies (1:500; Invitrogen, Waltham, MA, USA). The sections were washed in TBS before and after staining the cell nuclei with 4′,6-Diamidino-2-phenylindole dihydrochloride (DAPI, Sigma-Aldrich, Taufkirchen, Germany; 1:10,000 in TBS) and mounted with ProLong™ Gold Antifade Mountant (Invitrogen, Waltham, MA, USA). Staining of apoptotic cells was performed with a terminal deoxynucleotide transferase-mediated dUTP-X nick end labeling (TUNEL) method using the In Situ Cell Death Detection Kit (Roche, Penzberg, Germany) according to the manufacturers protocol.

### 2.4. Scoring of Colitis Severity

Scoring of colitis severity was performed on H&E-stained colon and terminal ileum tissue sections as previously described [23,24]. Briefly, the sum of the individual scores ((a) intestinal epithelial integrity (0–3), (b) extent of mucosal inflammation (0–3), and (c) submucosal pathological changes (0–3)) was calculated for each animal. The score for the colon was added to the score of the corresponding terminal ileum (=total score) and used as an indicator of disease severity.

### 2.5. Histological Quantification

Images of Nissl-stained sections were acquired at an Hamamatsu Nanozoomer S60 (Hamamatsu, Hamamatsu City, Japan), and further analysis was performed using the software NDP.view2 version 2.9.29 (Hamamatsu, Hamamatsu City, Japan). The thickness of the DG granule cell layer (GCL) was calculated from the mean thickness of the upper and lower GCL of the dorsal DG measured in 3–4 Nissl-stained slices per mouse.

Iba1^+^ myeloid cells and Iba1^+^Mcm2^+^-proliferating myeloid cells in the DG were quantified in at least three representative z stacks with 1 µm intervals and a total thickness of 20 µm per section and in all brain sections containing the DG of each mouse. The numbers of Iba1^+^ cells were normalized to the analyzed volume, resulting in a density of Iba1^+^ cells per mm^3^. The phagocytic state of ramified Iba1^+^ myeloid cells in the DG was quantified by scoring the cell morphology based on cellular Iba1 expression and the expression of the lysosomal protein CD68 according to Wilton et al. [25]. Overall, 50-207 Iba1^+^ cells were analyzed per mouse in at least three representative z stacks per section and in all brain sections containing the DG of each mouse. A total number of Iba1^+^ cells (*Casp8^fl^*/*Casp8^ΔIEC^*) in the DG of 14-week-old mice (740/717) and 24-week-old mice (1105/1018) was scored. Z stacks with 2 µm interval and a total thickness of 20 µm of sections co-stained for Iba1, Mcm2, and CD68 were acquired with an Axio Observer Z1 inverted fluorescence microscope with an ApoTome (Zeiss, Oberkochen, Germany) under a 40× objective.

To determine the total numbers of Mcm2^+^-proliferating cells in the subgranular zone of the DG and the numbers of DCX^+^ immature neurons, tile scans of confocal z stacks with 1 µm intervals and a total thickness of 20 µm were acquired at an LSM780 confocal laser scanning microscope with an inverted Axio Observer (Zeiss, Oberkochen, Germany) under a 20× objective, covering the whole DG on every brain section. All Mcm2^+^ and DCX^+^ cells were quantified, and obtained numbers were multiplied by 12. Moreover, all Mcm2^+^DCX^+^, Mcm2^+^NeuN^+^, and DCX^+^NeuN^+^ cells were counted and their percentages normalized to either all Mcm2^+^ or all DCX^+^ cells were calculated.

TUNEL^+^ apoptotic cells were quantified in 3–4 brain sections per animal and in 3 z stacks with 1 µm intervals recorded per section under a 20× objective using a LSM780 confocal laser scanning microscope with an inverted Axio Observer (Zeiss, Oberkochen, Germany). TUNEL^+^ cell density normalized to analyzed DG volumes was determined.

Histological quantifications were performed using the Zeiss Zen 2012 software (black edition, version 8.1.0.484; blue edition, version 1.1.2.0).

### 2.6. RNA Isolation and qPCR

RNA was isolated from snap-frozen forebrain, distal colon, ileum, and liver. RNA from colon, ileum, and liver was extracted using the NucleoSpin RNA Mini kit for RNA purification (Macherey-Nagel, Dueren, Germany). RNA from forebrain tissue was isolated by mechanically homogenizing the tissue in QIAzol reagent (Qiagen, Hilden, Germany) and using the RNeasy Mini Kit (Qiagen, Hilden, Germany). Reverse transcription was performed using the GoScript™ Reverse Transcription System (Promega, Walldorf, Germany) according to the manufacturers protocol. Quantitative real-time PCR was performed using the SsoFast™ EvaGreen^®^ Supermix (Bio-Rad, Feldkirchen, Germany) and the primers listed in Table 1. Samples were measured at a LightCycler^®^96 System with the LightCycler^®^ software (version 1.5.1; Roche, Penzberg, Germany). Quantification of gene expression was performed relative to the mean mRNA expression of the housekeeping genes *Hprt* and *Pgk1* using the ΔΔCt method.

### 2.7. Statistical Analysis

Statistical analyses were performed using GraphPad Prism 10 (Dotmatics, Boston, MA, USA) and R (version 4.4.3), RStudio (version 2024.12.1). Data were tested for normal distribution using a Shapiro–Wilk test. Colitis scores and histological data were further tested for statistically significant differences between genotypes or ages using two-way analysis of variance (ANOVA) followed by a Tukey’s multiple comparisons test. In TUNEL^+^ cell density data, the groups of 24-week-old *Casp8^fl^* and *Casp8^ΔIEC^* mice each contained one outlier exceeding all other values by more than 3-fold. The outliers were confirmed using the ROUT method [26] in GraphPad Prism and the Gubbs test in R and were excluded from further analyses. Microglial activation scores based on morphology and CD68 expression were analyzed via multiple paired *t*-tests as previously described [25]. Gene expression values derived from qPCR experiments were compared separately for 14-week-old and 24-week old cohorts between *Casp8^fl^* and *Casp8^ΔIEC^* mice via unpaired *t*-tests. The association between microglial parameters and “Colitis+Ileitis Score” was analyzed using a Pearson’s coefficient of correlation analysis using R. Results with a probability level of *p* < 0.05 were considered statistically significant. All data are presented as mean ± standard deviation; *ns p* > 0.05, * *p* ≤ 0.05, ** *p* ≤ 0.01, *** *p* ≤ 0.001.

The graphical abstract was created in BioRender. Masanetz, R. (2025) https://BioRender.com/si6b2qp.

## 3. Results

### 3.1. Casp8^ΔIEC^ Mice Develop Ileocolitis and Extraintestinal Inflammation

In this study, *Casp8^ΔIEC^* mice were used to determine the effects of gut inflammation on the microglial compartment and adult neurogenesis in the hippocampal DG. The genetically modified *Casp8^ΔIEC^* mouse model of IBD spontaneously develops inflammation in the gastrointestinal tract, specifically in the terminal ileum and colon [19,20] depending on the microbiome, and displays a mild inflammatory phenotype in the liver [21,27]. Mice with ileocolitis (*Casp8^ΔIEC^*) were compared to control mice (*Casp8^fl^*) at the age of 14 and 24 weeks, respectively (Figure 1A). Histologic scoring of H&E-stained colon and terminal ileum tissue sections revealed moderate colitis and ileitis in 14- and 24-week-old *Casp8^ΔIEC^* mice compared to *Casp8^fl^* mice (*Casp8^fl^* vs. *Casp8^ΔIEC^* Colon + Ileum 14 weeks: 0.8 ± 0.3 vs. 7.8 ± 1.6, *p* < 0.001; 24 weeks: 1.1 ± 0.5 vs. 11.1 ± 2.9, *p* < 0.001; Colon 14 weeks: 0.6 ± 0.2 vs. 3.1 ± 1.6, *p* = 0.024; 24 weeks: 0.6 ± 0.5 vs. 5.6 ± 2.8, *p* < 0.001; Ileum 14 weeks: 0.2 ± 0.3 vs. 4.8 ± 1.1, *p* < 0.001; 24 weeks: 0.5 ± 0.4 vs. 5.4 ± 2.9, *p* < 0.001; Figure 1C). Besides colitis and ileitis, 14- and 24-week-old *Casp8^ΔIEC^* mice displayed mild liver inflammation, as indicated by increased mRNA levels of the inflammatory genes tumor necrosis factor (Tnf; *Casp8^fl^* vs. *Casp8^ΔIEC^* 14 weeks: 1.0 ± 0.6 vs. 3.8 ± 3.1, *p* = 0.061; 24 weeks: 1.0 ± 0.6 vs. 2.3 ± 0.3, *p* = 0.002; Figure 1D) and lipocalin 2 (Lcn2; *Casp8^fl^* vs. *Casp8^ΔIEC^* 14 weeks: 1.0 ± 0.6 vs. 131.8 ± 77.3, *p* = 0.002; 24 weeks: 1.0 ± 0.4 vs. 405.2 ± 327.1, *p* = 0.014; Figure 1D) but not interleukin 1 beta (Il1b; Casp8^fl^ vs. Casp8^ΔIEC^ 14 weeks: 1.0 ± 0.9 vs. 1.7 ± 0.8, p=0.198; 24 weeks: 1.0 ± 1.4 vs. 0.7 ± 0.3, *p* = 0.681; Figure 1D), pointing towards systemic inflammation.

### 3.2. Microglial State in the Dentate Gyrus Is Maintained in Casp8^ΔIEC^ Mice

Microglia are highly plastic brain-resident macrophages responsive to chronic peripheral inflammation [28,29] and involved in the regulation of adult hippocampal neurogenesis [12,16]. We therefore hypothesized that chronic ileocolitis in the *Casp8^ΔIEC^* mouse model of CD results in microglia activation and impairment of adult neurogenesis in the hippocampal DG. In the DG of *Casp8^ΔIEC^* mice, the cell density of ramified Iba1^+^ myeloid cells (*Casp8^fl^* vs. *Casp8^ΔIEC^* 14 weeks: 9738 ± 1341 mm^3^ vs. 9372 ± 886 mm^3^, *p* = 0.974; 24 weeks: 11,104 ± 2270 mm^3^ vs. 9679 ± 1282 mm^3^, *p* = 0.147; Figure 2A) and the percentage of proliferating Iba1^+^Mcm2^+^ double positive cells among all Iba1^+^ cells (*Casp8^fl^* vs. *Casp8^ΔIEC^* 14 weeks: 8.5 ± 3.6% vs. 17.6 ± 15.7%, *p* = 0.230; 24 weeks: 11.8 ± 2.9% vs. 8.4 ± 4.6%, *p* = 0.839; Figure 2A) were comparable to *Casp8^fl^* mice in both age groups. The activation of microglia was scored as previously described [25] based on the morphology and expression of the lysosomal protein CD68, reflecting the level of microglial phagocytic activity. In 14- and 24-week-old *Casp8^ΔIEC^* mice, the phagocytic phenotype of DG microglia was comparable to *Casp8^fl^* control mice (*Casp8^fl^* vs. *Casp8^ΔIEC^* 14 weeks, stage 0: 29.3 ± 11.6% vs. 18.0 ± 10.7%, *p* = 0.286; stage 1: 27.3 ± 9.4% vs. 27.2 ± 4.7%, *p* > 0.999; stage 2: 33.7 ± 15.2% vs. 37.5 ± 11.2%, *p* = 0.954; stage 3: 7.9 ± 3.7% vs. 13.3 ± 6.0%, *p* = 0.286; stage 4: 0.8 ± 0.8% vs. 3.3 ± 3.8%, *p* = 0.286; stage 5: 0.7 ± 1.5% vs. 0.7 ± 0.7%, *p* > 0.999; 24 weeks, stage 0: 21.8 ± 6.6% vs. 17.9 ± 7.9%, *p* = 0.958; stage 1: 23.4 ± 7.2% vs. 23.1 ± 4.6%, *p* = 0.995; stage 2: 39.8 ± 12.3% vs. 44.4 ± 14.4%, *p* = 0.958; stage 3: 12.9 ± 4.1% vs. 12.1 ± 8.7%, *p* = 0.995; stage 4: 1.4 ± 1.0% vs. 1.4 ± 2.1%, *p* = 0.995; stage 5: 0.6 ± 0.9% vs. 0.9 ± 1.0%, *p* = 0.958; Figure 2B), indicating that there is no overt inflammatory activation of microglia. An absent microglial response to gastrointestinal inflammatory cues in *Casp8^ΔIEC^* mice was supported by a lacking correlation with histological scores of colitis and ileitis within individual animals (Appendix A). This is corroborated by the mRNA expression analysis of genes associated with microglia homeostasis (*purinergic receptor P2Y12*, *P2ry12*; *Casp8^fl^* vs. *Casp8^ΔIEC^* 14 weeks: 1.0 ± 0.1 vs. 1.1 ± 0.2, *p* = 0.646; 24 weeks: *Casp8^fl^*: 1.0 ± 0.2 vs. 0.9 ± 0.4, *p = 0.569*; Figure 2C) and activation (*apolipoprotein E*, *Apoe*; *Casp8^fl^* vs. *Casp8^ΔIEC^* 14 weeks: 1.0 ± 0.4 vs. 1.2 ± 0.4, *p* = 0.611; 24 weeks: 1.0 ± 0.3 vs. 0.9 ± 0.6, *p* = 0.707; Figure 2C) in the forebrain, which revealed comparable expression among groups. Moreover, the gene expression of the tight junction proteins claudin 5 (*Cldn5*; *Casp8^fl^* vs. *Casp8^ΔIEC^* 14 weeks: 1.0 ± 0.2 vs. 1.0 ± 0.2, *p* = 0.705; 24 weeks: 1.0 ± 0.2 vs. 0.8 ± 0.3, *p* = 0.379; Figure 2C) and zona occludens 1 (ZO-1; *tight junction protein 1*, *Tjp1*; *Casp8^fl^* vs. *Casp8^ΔIEC^* 14 weeks: 1.0 ± 0.3 vs. 1.1 ± 0.4, *p* = 0.728; 24 weeks: 1.0 ± 0.3 vs. 0.8 ± 0.4, *p* = 0.389; Figure 2C), which are critical for blood–brain barrier integrity, were similar between groups, indicating no major blood–brain barrier disruption. These findings indicate that there is no major inflammatory microglial response or blood–brain barrier leakage in the DG of *Casp8^ΔIEC^* mice.

### 3.3. Adult Neurogenesis in the Dentate Gyrus Is Preserved Despite Ileocolitis

During adulthood, new neurons are born and mature in the subgranular zone (neurogenic niche) of the DG before integrating into the GCL. This process is susceptible to regulation by inflammatory factors derived from microglia [30]. In the *Casp8^ΔIEC^* mouse model, the thickness of the DG GCL was comparable between groups (Casp8^fl^ vs. Casp8^ΔIEC^ 14 weeks: 65.6 ± 5.3 µm vs. 64.2 ± 4.6 µm, *p* = 0.970; 24 weeks: 65.4 ± 6.2 µm vs. 67.4 ± 7.2 µm, *p* = 0.999; Figure 3A), indicating no major loss of GCL neurons or degeneration of the DG. Moreover, the absolute numbers of proliferating Mcm2^+^ cells in the subgranular zone of the DG did not significantly differ between groups (Casp8^fl^ vs. Casp8^ΔIEC^ 14 weeks: 4328 ± 1911 vs. 3934 ± 1013, *p* = 0.960; 24 weeks: 3626 ± 1146 vs. 3748 ± 3427, *p* > 0.999; Figure 3C). Besides absolute numbers, beginning neuronal differentiation of proliferating Mcm2^+^ progenitor cells indicated by the percentage of expression of the immature neuroblast marker DCX was not altered by ileocolitis (Casp8^fl^ vs. Casp8^ΔIEC^ 14 weeks: 70.9 ± 2.6% vs. 68.4 ± 5.6%, *p* = 0.923; 24 weeks: 59.6 ± 8.5% vs. 61.0 ± 6.9%, *p* > 0.999; Figure 3C). To address the further maturation of neural progenitor cells during experimental ileocolitis, we characterized DCX^+^ neuroblasts in the DG and found no significant changes in absolute numbers of DCX^+^ cells between *Casp8^ΔIEC^* mice and *Casp8^fl^* controls (Casp8^fl^ vs. Casp8^ΔIEC^ 14 weeks: 10,660 ± 4516 vs. 10,538 ± 3112, *p* > 0.999; 24 weeks: 8328 ± 4380 vs. 8013 ± 7236, *p* = 0.957; Figure 3D). A distinct co-expression analysis to compare the ratios of rather early DCX^+^Mcm2^+^ neuroblasts revealed no changes between groups (Casp8^fl^ vs. Casp8^ΔIEC^ 14 weeks: 28.9 ± 2.4% vs. 25.7 ± 1.8%, *p* = 0.695; 24 weeks: 28.2 ± 6.9% vs. 28.0 ± 4.5%, *p* = 0.989; Figure 3D), while the percentages of more mature DCX^+^NeuN^+^ neuroblasts showed a minor, but significant, increase in 24-week-old *Casp8^fl^* control mice compared to younger *Casp8^fl^* mice, and age-matched *Casp8^ΔIEC^* mice with ileocolitis (Casp8^fl^ vs. Casp8^ΔIEC^ 14 weeks: 18.1 ± 3.2% vs. 17.9 ± 2.1%, *p* = 0.996; 24 weeks: 22.3 ± 3.7% vs. 19.5 ± 3.7%, *p* = 0.037; Figure 3D) revealed no major shift in maturation between all four experimental groups. Furthermore, the density of TUNEL^+^ apoptotic cells were comparable between diseased *Casp8^ΔIEC^* and healthy *Casp8^fl^* mice at both ages (Casp8^fl^ vs. Casp8^ΔIEC^ 14 weeks: 1489 ± 1020 mm^3^ vs. 1090 ± 521 mm^3^, *p* > 0.999; 24 weeks: 1867 ± 1643 mm^3^ vs. 994 ± 583 mm^3^, *p* = 0.907; Figure 3E), indicating no changes in cell death during the neurogenic process between the groups. Finally, we assessed gene expression of neurotrophic factors supporting adult hippocampal neurogenesis. In line with the maintained stages of neurogenesis detected by histology, we did not observe alterations in the expression of *Bdnf* (Casp8^fl^ vs. Casp8^ΔIEC^ 14 weeks: 1.0 ± 0.6 vs. 0.9 ± 0.3, *p* = 0.673; 24 weeks: 1.0 ± 0.3 vs. 0.8 ± 0.4, *p* = 0.472; Figure 3F), *Gdnf* (Casp8^fl^ vs. Casp8^ΔIEC^ 14 weeks: 1.0 ± 1.0 vs. 0.8 ± 0.5, *p* = 0.665; 24 weeks: 1.0 ± 0.7 vs. 1.1 ± 1.2, *p* = 0.920; Figure 3F), and *Igf1* (Casp8^fl^ vs. Casp8^ΔIEC^ 14 weeks: 1.0 ± 1.0 vs. 1.0 ± 0.6, *p* = 0.966; 24 weeks: 1.0 ± 0.4 vs. 0.9 ± 0.5, *p* = 0.686; Figure 3F) between mice with ileocolitis and controls in both age cohorts. In conclusion, adult hippocampal neurogenesis, including proliferation, maturation, apoptotic cell death, and expression of neurotrophic factors, is not altered in the presence of ileocolitis in *Casp8^ΔIEC^* mice.

## 4. Discussion

IBD poses a rising global health care burden [31,32,33,34]. Besides chronic intestinal inflammation, extraintestinal manifestations have been observed in nearly half of all IBD patients [35] and are more prevalent among CD patients [35,36]. Extraintestinal manifestations in patients diagnosed with IBD encompass the nervous system [37,38]. Specifically, IBD patients frequently exhibit neuropsychiatric symptoms, including depression and anxiety, which often correlate with inflammatory disease activity [1]. Thus, neuropsychiatric symptoms in IBD may be mediated by the propagation of gut-derived inflammation to the brain.

The *Casp8^ΔIEC^* mice utilized in this study exhibited spontaneous development of moderate ileocolitis, with some variation in the disease severity based on the histologic scoring. In addition to colitis and ileitis, *Casp8^ΔIEC^* mice demonstrated an increased expression of the inflammatory genes *Tnf* and *Lcn2* in the liver, suggesting mild hepatic inflammation as previously reported [21,27]. In recent years, the interaction between the gut, the liver, and the brain—the gut–liver–brain axis—has gained tremendous research attention [39].

In light of these findings and in alignment with previous studies indicating compromised adult hippocampal neurogenesis in the context of experimental colitis [28,40,41], we hypothesized that the peripheral inflammation in the IEC-specific *Casp8^ΔIEC^* mouse model leads to microglia activation and subsequently to an impaired adult hippocampal neurogenesis. The process of adult hippocampal neurogenesis in the DG is influenced by various immune components, including microglia cells, which regulate cell proliferation, differentiation, and survival of newborn neurons and can act proneurogenically, depending on the inflammatory microenvironment [15,16,42]. In the brain, microglia account for the largest immune cell population. These parenchymal macrophages rapidly respond to changes in the tissue environment, as induced by external stimuli, such as inflammation [43,44]. The impact of inflammation on the highly plastic process of hippocampal neurogenesis is evident, as pharmacologically blocking peripherally induced brain inflammation by treatment with the nonsteroidal anti-inflammatory drug (NSAID) indomethacin restored hippocampal neurogenesis in mice [14].

We thus investigated the impact of chronic ileocolitis in the *Casp8^ΔIEC^* mouse model on the myeloid cell compartment and adult neurogenic niche of the hippocampal DG. Our results demonstrate that, despite peripheral inflammation, in both the adolescent (14-week-old) and adult (24-week-old) *Casp8^ΔIEC^* mice, signs of inflammatory microglial activation and blood–brain barrier leakage in the DG were not detected. Furthermore, adult hippocampal neurogenesis is maintained despite chronic gut-derived peripheral inflammation in *Casp8^ΔIEC^* mice. The detected minimal decrease in NeuN expression in DCX^+^ neuroblasts in 24-week-old *Casp8^ΔIEC^* mice compared to *Casp8^fl^* controls can mainly be attributed to an unexpected slight increase in proportional NeuN-positivity in *Casp8^fl^* mice between 14 and 24 weeks of age but is very unlikely to reflect a biologically relevant impairment in neuroblast maturation due to ileocolitis, especially as all other neurogenesis-related parameters are preserved. This indicates that not only microglia but also other cellular and molecular drivers of neuroinflammation, as well as microbial factors previously described in IBD [4,45], are unable to trigger an overt microglial response and substantially impair adult neurogenesis in the DG of *Casp8^ΔIEC^* mice. Previous studies employing the DSS-induced colitis model have reported conflicting results regarding microglial density and activation state in the hippocampus [28,40,41,46]. While an increased density of microglia and microglial activation were only observed in the hippocampus of acute DSS-induced colitis mice [28,40,46], elevated inflammatory cytokine expression was detected in the hippocampus of both acute and chronic DSS-induced colitis mice [28,47], whereas the density of microglia was not altered in the hippocampus of chronic DSS-induced colitis compared to control mice [41].

Moreover, conflicting results have been reported regarding the effect of DSS-induced colitis on the generation and survival of new neurons in the hippocampal DG [28,40,41]. In acute DSS-induced colitis mice, Vitali et al. [41] observed reduced densities of glial fibrillary acidic protein (Gfap)-expressing radial glia-like stem cells and SRY-box transcription factor 2 (Sox2)-expressing intermediate progenitor cells with no changes in the densities of Ki67^+^-proliferating cells and DCX^+^ immature neurons in the hippocampal DG of 8-week-old male C57BL/6 mice fed with 0.7% DSS for 7 days plus 7 days recovery, whereas Takahashi et al. [40] reported significantly reduced numbers of 5-Bromo-2′-Desoxyuridine (BrdU)-positive proliferating cells and DCX^+^ immature neurons in the DG in 6-7-week-old male ddY mice fed with 1.5% DSS for 7 days compared to control mice. In contrast, Gampierakis et al. [28] found significantly increased densities of Ki67^+^-proliferating cells, Sox2^+^ intermediate progenitor cells, and DCX^+^ immature neurons in the DG of 2.5-month-old male C57BL/6 mice fed with 2.5% DSS for 7 days compared to control mice. In chronic DSS-induced colitis, Zonis et al. [48] observed reduced numbers of Sox2^+^ and DCX^+^ cells in the DG subgranular zone, which is in line with the reduced hippocampal gene and protein expression of *Sox2*, *DCX*, and DCX reported by Barnes et al. [47]. In contrast, Vitali et al. [41] observed no difference in the densities of Gfap^+^ radial glia-like stem cells, Sox2^+^, and DCX^+^ cells but a reduced density of Ki67^+^-proliferating cells in the hippocampus of chronic DSS-induced colitis mice. The reported findings do not imply a dose-dependent effect, and the heterogeneous results may be attributed to the differences in the experimental design paradigm, age, and genetic background of animals, disease severity, and the resulting systemic inflammatory response.

Thus, we employed the IEC-specific *Casp8^ΔIEC^* transgenic mouse model to overcome the limitation of the highly variable and less specific colitis induction applied in previous studies employing the DSS-induced colitis model. While numbers of proliferating progenitor cells and maturating neuroblasts as well as apoptosis in the DG of *Casp8^ΔIEC^* mice were comparable to those in the DG of *Casp8^fl^* mice, minimal changes in the expression of NeuN within maturating DCX^+^ neuroblasts are possible, as mice age and ileocolitis progresses. In the light of the high vulnerability of adult hippocampal neurogenesis and previous findings in DSS-treated rodents, our finding of largely maintained adult hippocampal neurogenesis may appear surprising, but preserved adult hippocampal neurogenesis has previously been described in other models of even more severe chronic peripheral inflammation [22,49]. As peripheral inflammation was shown to affect the brain in a regionally different manner [29], the present findings do not allow the conclusion of the overall cerebral resilience to ileocolitis. Future studies therefore must compare regional immune responses in the brain of the *Casp8^ΔIEC^* model. This may include further brain regions which might be involved in neuropsychiatric comorbidity of IBD independent of the hippocampus. Moreover, an impairment of adult hippocampal neurogenesis may be restricted to certain subtypes of systemic inflammation and may be predicted by more thorough immunophenotyping of ileocolitis and systemic inflammation in IBD patients in the future.

## 5. Conclusions

In summary, hippocampal neurogenesis and microglial response in the DG are largely unaffected by ileocolitis in the *Casp8^ΔIEC^* transgenic mouse model of CD.

## Figures and Tables

**Figure 1 cells-14-00841-f001:**
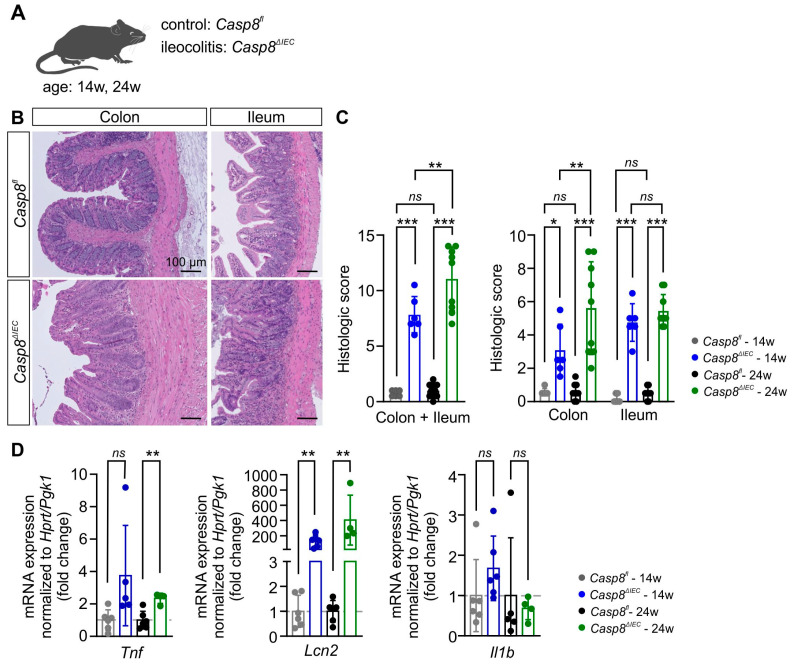
Ileocolitis in the *Casp8^ΔIEC^* mouse model of inflammatory bowel disease. (**A**) *Casp8^ΔIEC^* mice and *Casp8^fl^* control mice aged 14 and 24 weeks were used. (**B**) Representative H&E images of colon and terminal ileum tissue sections of 24-week-old *Casp8^ΔIEC^* and *Casp8^fl^* mice. Scale bar: 100 µm. (**C**) The combined and the individual histologic scores of the colon and ileum were significantly higher in 14- and 24-week-old *Casp8^ΔIEC^* mice (14 weeks: *n* = 6, 3 female/3 male; 24: weeks: *n* = 9, 6 female/3 male) compared to age-matched *Casp8^fl^* mice (14 weeks: *n* = 6, 3 female/3 male; 24: weeks: *n* = 9, 6 female/3 male). (**D**) Gene expression analysis revealed increased mRNA levels of *Tnf* and *Lcn2* but not *Il1b* in the liver of 14- and 24-week-old *Casp8^ΔIEC^* mice (14 weeks: *n* = 6, 3 female/3 male; 24: weeks: *n* = 4, 4 female/0 male) compared to age-matched *Casp8^fl^* mice (14 weeks: *n* = 6, 3 female/3 male; 24: weeks: *n* = 6, 6 female/0 male). Significance in (**C**) was tested by two-way ANOVA followed by Tukey’s post hoc test. Significance in (**D**) was tested by unpaired *t*-tests. Numbers above brackets indicate p-values for pairwise comparisons. All data are shown as mean ± standard deviation; *ns p* > 0.05, * *p* ≤ 0.05, ** *p* ≤ 0.01, *** *p* ≤ 0.001.

**Figure 2 cells-14-00841-f002:**
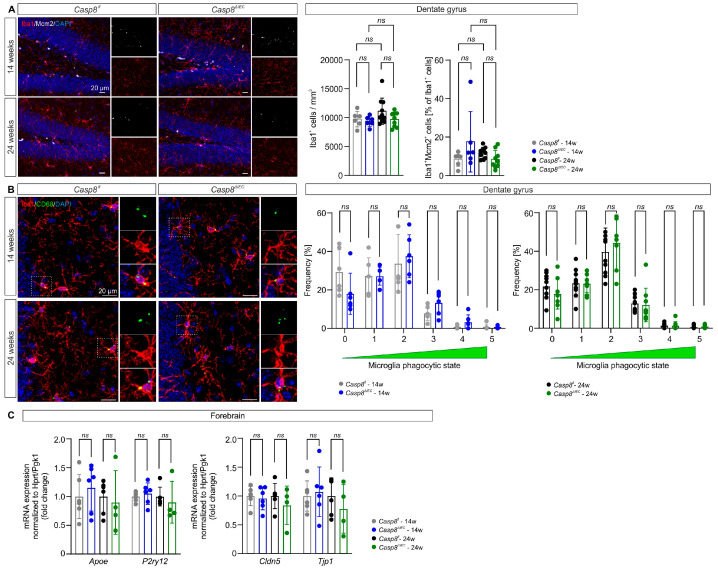
Microglia proliferation and activation in the DG. (**A**) The densities of ramified Iba1^+^ myeloid cells (microglia) and Iba1^+^Mcm2^+^-proliferating microglia were similar between groups. (**B**) Similar number of phagocytic microglia scored by cell morphology and CD68 expression were observed in *Casp8^ΔIEC^* and *Casp8^fl^* mice in both age groups. (**C**) In the forebrain, expression of genes associated with microglia activation (*Apoe*) and homeostasis (*P2ry12*) and of genes encoding tight junction proteins (*Cldn5*, *Tjp1*) was similar in all groups. Data in (**A**,**B**) are representative of *n* = 6 (3 female/3 male) 14-week-old and *n* = 9 (6 female/3 male) 24-week-old *Casp8^ΔIEC^* mice as well as *n* = 6 (3 female/3 male) 14-week-old and *n* = 10 (7 female/3 male) 24-week-old *Casp8^fl^* controls and were tested for significance using two-way ANOVA followed by Tukey’s post hoc test ((**A**), Iba1^+^ cells), two-way ANOVA followed by Tukey’s post hoc test ((**A**), Iba1^+^Mcm2^+^ cells), or multiple unpaired *t*-tests (**B**), respectively. Data in (**C**) are representative of *n* = 6 (3 female/ 3 male) 14-week-old and *n* = 4 (4 female/0 male) 24-week-old *Casp8^ΔIEC^* mice as well as *n* = 6 (3 female/3 male) 14-week-old and *n* = 6 (6 female/0 male) 24-week-old *Casp8^fl^* controls and were tested for significance by unpaired *t*-tests. All data are shown as mean ± standard deviation; *ns p* > 0.05.

**Figure 3 cells-14-00841-f003:**
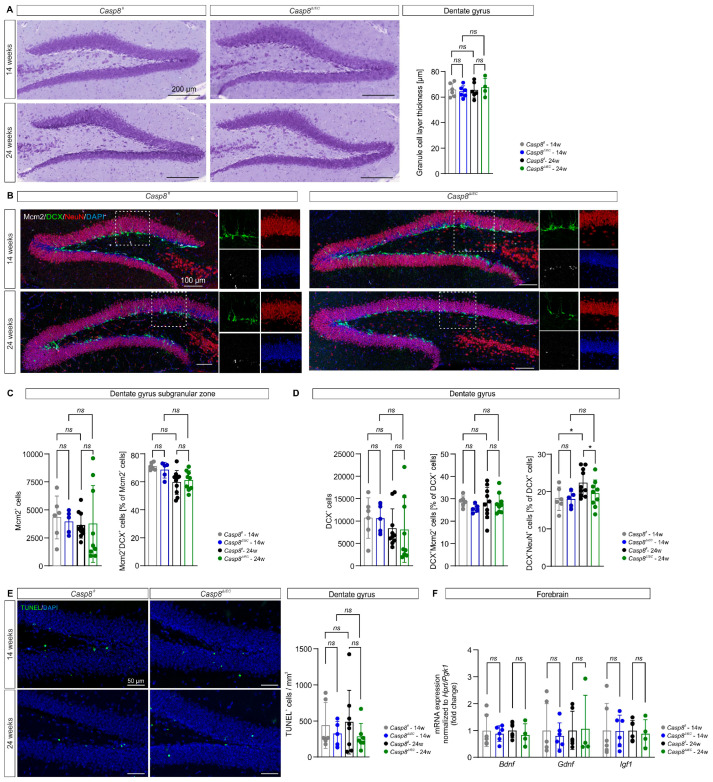
Adult hippocampal neurogenesis. (**A**) The thickness of the DG GCL was similar in all groups. Scale bar: 200 µm. (**B**) Representative images showing immunofluorescent co-staining of the DG for Mcm2 (white), DCX (green), NeuN (red), and DAPI (blue) in 14- and 24-week-old *Casp8^fl^* and *Casp8^ΔIEC^* mice. Scale bar: 100 µm. (**C**) Quantification of total proliferating Mcm2^+^ cells and the percentage of Mcm2^+^ cells co-expressing DCX showing no difference between mice with ileocolitis and controls. (**D**) Quantification of total maturating DCX^+^ neuroblasts and the percentages of DCX^+^ cells co-expressing Mcm2 or NeuN, respectively, showing a slight increase in DCX^+^NeuN^+^ cells in 24-week-old *Casp8^fl^* mice compared to 14-week-old Casp8*^fl^* and 24-week-old *Casp8^ΔIEC^* mice. (**E**) Densities of TUNEL^+^ apoptotic cells per mm^3^ were comparable between *Casp8^ΔIEC^* and *Casp8^fl^* mice in both age groups. (**F**) Expression of genes encoding the trophic factors *Bdnf*, *Gdnf*, and *Igf1* in the forebrain was similar in 14- and 24-week-old *Casp8^ΔIEC^* mice compared to age-matched *Casp8^fl^* mice. Data in (**A**,**F**) represent *n* = 6 (3 female/3 male) 14-week-old and *n* = 4 (4 female/0 male) 24-week-old *Casp8^ΔIEC^* mice as well as *n* = 6 (3 female/3 male) 14-week-old and *n* = 6 (6 female/0 male) 24-week-old *Casp8^fl^* controls. Data in (**C**,**D**) represent *n* = 5 (3 female/2 male) 14-week-old and *n* = 9 (6 female/3 male) 24-week-old *Casp8^ΔIEC^* mice as well as *n* = 6 (3 female/3 male) 14-week-old and *n* = 10 (7 female/3 male) 24-week-old *Casp8^fl^* controls. Data in (**E**) represent *n* = 6 (3 female/3 male) 14-week-old and *n* = 7 (5 female/2 male) 24-week-old *Casp8^ΔIEC^* mice as well as *n* = 6 (3 female/3 male) 14-week-old and *n* = 8 (5 female/3 male) 24-week-old *Casp8^fl^* controls. Data in (**A**,**C**–**E**) were tested for significant differences between genotypes and ages by 2way ANOVA followed by Tukey’s post hoc test. Data in (**F**) were tested for significant differences between genotypes via unpaired *t*-tests. All data are shown as means ± standard deviation; *ns p* > 0.05, * *p* ≤ 0.05.

**Table 1 cells-14-00841-t001:** Sequences of primers used for qPCR.

Gene	Sequence Forward Primer (5′-3′)	Sequence Reverse Primer (5′-3′)
*Apoe*	GAACAACCCGCCTCGTGA	AGCTCCTTCCGAAACAAGTCC
*Bdnf*	GCTCATCTTTGCCAGAGCCC	AGCAGCTTTCTCAACGCCT
*Cldn5*	GTTAAGGCACGGGTAGCACT	TACTTCTGTGACACCGGCAC
*Gdnf*	ACCCTGCTAGAAAACGCGAG	ACGGAGATCCGGGCAAAAG
*Hprt*	GTCATGTCGACCCTCAGTCC	GCAAGTCTTTCAGTCCTGTCC
*Igf1*	ATACAGCCAACGGGAAACAG	CAACAAAGCTGGATGCCTGTC
*Il1b*	TGACAGTGATGAGAATGACCTG	CCACGGGAAAGACACAGGTA
*Lcn2*	ACGGACTACAACCAGTTCGC	ATGCATTGGTCGGTGGGG
*P2ry12*	AGTGCAAGAACACTCAAGGC	GACGGTGTACAGCAATGGGA
*Pgk1*	GTCGTGATGAGGGTGGACTT	AACGGACTTGGCTCCATTGT
*Tnf*	TAGCCCACGTCGTAGCAAAC	GCAGCCTTGTCCCTTGAAGA
*Tjp1*	GGAGATGTTTATGCGGACGG	CCATTGCTGTGCTCTTAGCG

## Data Availability

The original contributions presented in this study are included in the article. Further inquiries can be directed to the corresponding author.

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
