# Peer review of "Absence of Microglial Activation and Maintained Hippocampal Neurogenesis in a Transgenic Mouse Model of Crohn’s Disease"

_cells, 2025, doi:10.3390/cells14110841_

Round 1
Reviewer 1 Report
Comments and Suggestions for Authors
Cells
Resilience of microglial response and hippocampal neurogenesis in a transgenic mouse model of Crohn’s Disease
Masanetz et al.
Crohn's disease (CD) is characterized by an inflammation of the digestive tract which is thought to involve an overactive immune response to normal gut bacteria in genetically susceptible people. Recent studies with focus on the gut-brain axis center on the role of the brain’s endogenous immune cells, microglia. Microglia contribute to neuroplasticity of the hippocampus under physiological conditions and orchestrate the acute inflammatory response to pathogenic stimuli. The current long-term study aimed to explore microglial cells and adult neurogenesis in a transgenic mouse model “that spontaneously develops ileocolitis” therefore mimicking CD. Histology and gene expression data for pro-inflammatory markers confirm chronic ileocolitis in Casp8AlEC mice. Immuno-data for the adult brain reveal no alterations in cell proliferation, microglial numbers or apoptosis in the dentate gyrus between transgenic mice. Although the study is very interesting and results are surprising, the provided data not fully proof the manuscripts conclusion, lack data on adult neurogenesis, and an in-depth analysis and discussion. Many questions remain on i) whether there are sex differences since male and female were used, ii) what is the age effect because it is a chronic disease, iii) how can the different/counteracting functions of microglia be distinguished to point out ‘resilience’, and what are possible compensatory mechanisms – or was the hypothesis wrong? The title should be adjusted accordingly.
Please find my general comments, and specific ones on adult neurogenesis, below:
- Please provide a few more sentences when animals spontaneously develop inflammation, and why the specific ages were chosen for the current study
- Please add numbers and p-values (up to 3 decimal p laces) into the text
- Specify animal numbers and sex used per specific group/staining, n=4 to 9/group is vague
- Immunohistochemistry: to obtain absolute numbers, i.e., of Mcm2+ and iba1+ cells, more slices (1:6 or 1:8) should be counted throughout the entire rostro-caudal extent of the dentate gyrus (DG) and all cells found within one DG per slice – usually by peroxidase staining which provides a better staining/more cells. The common procedure to determine cell proliferation and adult neurogenesis, would be to count the absolute number of proliferating cells (Mcm2+ in the current study by peroxidase staining) and label for co-expression/phenotypic analysis (i.e., Mcm2/iba1, Mcm2/Dcx etc.); to state that adult neurogenesis was determined, co-labeling of Mcm2/NeuN should be done.
- For iba1/Mcm2 co-expression, add a table with absolute numbers and percentages. Overall, edit the y-axes in Figure 2 and 3, there are often too large, while graphs are too small.
- M&M section: the authors write ‘sagittal slicing’, yet pictures refer to either coronal or horizontal slices? Figure 2 A and B both state 20 µm, however B shows a much higher magnification; overall, please add labeling to the images, define the different layers
- How was ‘density’ of cells measured, which is written throughout the manuscript?
- Line 206: “… indicating preserved microglial homeostasis (Figure 2 B)”; is having a ‘phagocytotic phenotype’ shown as frequency an indicator for homeostasis?
- Line 224/5 is misleading: cells are still born in adulthood, stem/progenitor cells proliferate in the subgranular zone, mainly become mature neurons while migrating into the granule cell layer (GCL, please check the spelling) and get incorporated into an existing network.
- Line 226 and beyond (Figure 3): regulation of aN by cytokines released by glial cells – i.e. microglia (?) – needs context and citations; the rational to measure GCL thickness is vague, integration of newly generated cells does not lead to a thicker layer, cell death is a compensatory factor; line 230: how was ‘survival’ measured?
Reviewer 2 Report
Comments and Suggestions for Authors
This is an interesting, well written, and well performed study. The authors should be commended on putting forward data that is largely negative; the scientific community has long demanded increased recognition of such studies. Furthermore, there is a paucity of mouse models of ileitis. Gut-brain axis work using mouse models of ileitis is desperately needed.
This study uses the, the Casp8 del IEC, mouse - a model of ileocolitis. Despite developing ileitis, colitis, and some systemic inflammation (as evidenced by increased Tnf and Lcn2 in the liver) there was no change in microglia in the DG or forebrain. There has been conflicting reports in the literature regarding microglial activation in response to intestinal inflammation. This report adds a unique insight using a model of ileocolitis.
The study is limited by a few key points. The authors acknowledge that a small sample is used. More subtle findings may be undetectable with smaller sample sizes. Furthermore, there is some concern about the generalizability of these findings. Are the results specific to this mouse model?
Here is a list of addressable points for the authors:
- The main critique of this study is the small sample size. This is particularly noticeable in figure 2 where several graphs look close to significance.
- Has a power analysis been performed for this data? Is the current data underpowered?
- By appearances, the data for 24w mice in 2b looks significant. The sample size may be too small to see statistical significance.
- This is a major concern. If the data is significant, this would by contrary to the claims made by the authors.
- The authors use an IEC specific Caspase 8 deletion model. This model was developed several years ago as seen in reference 19.
- In ref 25 (Environmental Microbial Factors Determine the Pattern of Inflammatory Lesions in a Murine Model of Crohn’s Disease–Like Inflammation), figure 1b shows colon histology is dependent on the microbiome (conventional vs spf). SPF mice had a score around 8.
- In ref 21 (Intestinal epithelial Caspase-8 signaling is essential to prevent necroptosis during Salmonella Typhimurium induced enteritis), figure 2f shows that colon inflammation of uninfected casp8 del mice was around 2.
- In the present study, colon histology averages ~3 at 14 weeks and ~6 at 24 weeks. There does appear to be some variability in the colitis exhibited by these mice. Can the authors account for this? Is there any correlation between severity of ileitis/colitis and the later microglial parameters?
- How many outliers were removed using ROUT and did removing outliers alter the results? Was there a justification outside of identification with ROUT for removing outliers? Is there a policy that is generally followed by the group to remove outliers or is the decision made ad hoc?
- Per the GraphPad documentation, "...ROUT tests assume that all the values are sampled from a Gaussian distribution, with the possible exception of one (or a few) outliers from a different distribution. If the underlying distribution is not Gaussian, then the results of the outlier test is unreliable."
- Furthermore, nonparametric tests were used in all figures. Nonparametric tests are generally more robust to outliers so removing outliers has less justification.
- Please add the primer sequence used for Lcn2.
- The primer sequences for Hprt, Ikbkb, and Pgk1 are included in methods but are not used in any figures.
Round 2
Reviewer 1 Report
Comments and Suggestions for Authors
The manuscript has much improved, and the outcome is clear. However, the layout and data presentation need additional work. Please take out p values from inside the figures which are not significant; it is confusing and not relevant; usually you add * to the figures, and p-values to the text, there, the corresponding figure is always added inside the same brackets. Furthermore, there should be no space between the number and the unit ‘%’; also use ‘,’ instead of multiple ‘:’ when defining the groups, i.e., lines 253, and throughout 3.3. Lines 251 to 259 should be rewritten; the word frequency is missing and the 5 stages; using a comparison instead of a list is also more appropriate, i.e., (Casp8x vs. Casp8y 14 weeks, stage 0: 29.3 ± 11.6% vs 18.0 ± 10.7%, 1: ….; 24 weeks, stage 0: …; Figure 2B). Similar line 293 and further, use ‘vs’ when comparing the groups, instead of listing each group one after the other.
I also think the dot plots figures before were better and consider taking out outliers; often the SD is 100% of the value which is unusual (for example in Figure B, the green graph shows 4000 cells and the SD reveals 7500). Overall, SEM is often used instead of SD. Figure 3D, 3rd graph’s y-axis should read DCX/NeuN not NeuN/DCX.
Author Response
Reviewer 1
The manuscript has much improved, and the outcome is clear. However, the layout and data presentation need additional work.
Comment:
Please take out p values from inside the figures which are not significant; it is confusing and not relevant; usually you add * to the figures, and p-values to the text, there, the corresponding figure is always added inside the same brackets.
Answer:
We thank the reviewer for this comment and have replaced non-significant p-values (p > 0.05) by ‘ns’ and used ‘*’, ‘**’, and ‘***’ for significant p-values, i.e. p ≤ 0.05, p ≤ 0.01, and p ≤ 0.001, respectively. The text has been adjusted accordingly and corresponding figures were added to the brackets of mentioned p-values.
Comment:
Furthermore, there should be no space between the number and the unit ‘%’; also use ‘,’ instead of multiple ‘:’ when defining the groups, i.e., lines 253, and throughout 3.3. Lines 251 to 259 should be rewritten; the word frequency is missing and the 5 stages; using a comparison instead of a list is also more appropriate, i.e., (Casp8x vs. Casp8y 14 weeks, stage 0: 29.3 ± 11.6% vs 18.0 ± 10.7%, 1: ….; 24 weeks, stage 0: …; Figure 2B). Similar line 293 and further, use ‘vs’ when comparing the groups, instead of listing each group one after the other.
Answer:
We thank the reviewer for this helpful comment. We have reformatted the text according to the reviewer’s suggestion throughout section 3 to easily enable comparison between groups and to improve readability of the manuscript.
Comment:
I also think the dot plots figures before were better and consider taking out outliers; often the SD is 100% of the value which is unusual (for example in Figure B, the green graph shows 4000 cells and the SD reveals 7500). Overall, SEM is often used instead of SD.
Answer:
We thank the reviewer for this suggestion and have changed the figures to dot plots showing individual data points, thereby providing transparency of results. While standard error of the mean (SEM) is frequently used, standard deviation (SD) better visualizes variability within groups and dispersion of the individual values from the group mean and is considered the preferential measure for showing data variability (PMID: 23125963). We therefore decided to keep mean ± SD, even though error bars appear larger. Moreover, we thoroughly revised our statistical methods and rethought the justification of outlier removal after the initial comments of reviewer #2 and decided not to remove any further potential outlier as there were none detected by the applied tests in Prism and R. We are confident that our approach best adheres to the principles of good scientific practice.
Comment:
Figure 3D, 3rd graph’s y-axis should read DCX/NeuN not NeuN/DCX.
Answer:
We thank the reviewer for attentive reading and have corrected the y-axis label of Figure 3D.
Reviewer 2 Report
Comments and Suggestions for Authors
The authors have made significant changes and have improved the work substantially. Some minor points remain.
- The number of mice in figure legends is different from the values provided in the author's response letter. The number in the revised manuscript seems to be the same as the original manuscript. Is this a typo?
- Graphs were changed to simple bar charts instead of individual points from the original manuscript. While this is not a requirement, it is helpful to show the individual data points so readers can see how the data is distributed. Since each graph has a slightly different n, it is much easier to represent n with individual data points.
- Figure 2a has a slight typo: casp8fl 14 weeks vs casp8 del iec shows a p-value of 974
- Thank you for including the correlation data between intestinal inflammation and Iba1 cells. I would recommend including this as supplemental as it further strengthens the author's claims.
- Per the graphpad documentation, "Normality tests should not be used to automatically decide whether or not to use a nonparametric test." The decision to use a parametric or non-parametric test should be made a priori.
Author Response
Reviewer 2
The authors have made significant changes and have improved the work substantially. Some minor points remain.
Comment:
The number of mice in figure legends is different from the values provided in the author's response letter. The number in the revised manuscript seems to be the same as the original manuscript. Is this a typo?
Answer:
We thank the reviewer for raising this important point. As stated in the response letter of revision round 1, “We were lucky to obtain an additional cohort of 24-week-old Casp8ΔIEC mice and Casp8fl controls for histological analyses. For our main histological experiments to assess microglia (Fig. 2 A, B) and adult hippocampal neurogenesis (Fig. 3 B-E), sample sizes are now n=6 for both 14-week-old groups, n=9 for 24-week-old Casp8ΔIEC mice and n=10 for 24-week-old Casp8ΔIEC mice.” We very much regret the typo in the response letter which should be “[…] sample sizes are now n=6 for both 14-week-old groups, n=9 for 24-week-old Casp8ΔIEC mice and n=10 for 24-week-old Casp8fl control mice.” In the revised manuscript, animal numbers of individual analyses have been implemented into figure legends. Here, we very much regret the typo in the legend of Figure 2 which was corrected to match the actual sample size presented in Figure 2 A and B.
Comment:
Graphs were changed to simple bar charts instead of individual points from the original manuscript. While this is not a requirement, it is helpful to show the individual data points so readers can see how the data is distributed. Since each graph has a slightly different n, it is much easier to represent n with individual data points.
Answer:
We thank the reviewer for this suggestion and have changed the figures to dot plots showing individual data points, thereby providing transparency of results.
Comment:
Figure 2a has a slight typo: casp8fl 14 weeks vs casp8 del iec shows a p-value of 974
Answer:
We thank the reviewer for attentive reading. As suggested by Reviewer 1, we revised the figures have now replaced non-significant p-values (p > 0.05) by ‘ns’ and used ‘*’, ‘**’, and ‘***’ for significant p-values, i.e. p ≤ 0.05, p ≤ 0.01, and p ≤ 0.001, respectively. Exact p-values were included in the main text.
Comment:
Thank you for including the correlation data between intestinal inflammation and Iba1 cells. I would recommend including this as supplemental as it further strengthens the author's claims.
Answer:
We thank the reviewer for this suggestion and have included scatter plots showing the correlation of the sum colitis + ileitis score with all histological analyses related to microglia as Supplementary Figure S1. We have added relevant information in the methods and results section.
Comment:
Per the graphpad documentation, "Normality tests should not be used to automatically decide whether or not to use a nonparametric test." The decision to use a parametric or non-parametric test should be made a priori.
Answer:
We thank the reviewer for this comment. The full GraphPad documentation says that “Normality tests should not be used to automatically decide whether or not to use a nonparametric test. But they can help you make the decision.” We assumed and tested normal distribution of our continuous data. Previous recommendations on statistical testing of biological data have stated that parametric tests can also be applied to non-normally distributed data under certain circumstances (PMID: 29678516). In addition, applying ART ANOVA instead of 2way ANOVA as shown in the first round of revision did not change the results. We thus performed parametric tests for all datasets shown in this manuscript.